# Doxorubicin-Loaded Magnetic Nanoparticles: Enhancement of Doxorubicin's Effect on Breast Cancer Cells (MCF-7)

Elisa Parcero Hernandes [1] , Raquel Dosciatti Bini [2], Karina Midori Endo [1], Verci Alves de Oliveira Junior [1], Igor Vivian de Almeida [1], Gustavo Sanguino Dias [2] , Ivair Aparecido dos Santos [2], Paula Nunes de Oliveira [2] , Veronica Elisa Pimenta Vicentini [1] and Luiz Fernando Cotica [2,*]

[1] Department of Biotechnology, Genetics and Cell Biology, State University of Maringa, Av. Colombo, 5790, Maringa 87020900, PR, Brazil
[2] Department of Physics, State University of Maringa, Av. Colombo, 5790, Maringa 87020900, PR, Brazil
* Correspondence: lfcotica@dfi.uem.br

**Abstract:** The incidence of female breast cancer has increased; it is the most commonly diagnosed cancer, at 11.7% of the total, and has the fourth highest cancer-related mortality. Magnetic nanoparticles have been used as carriers to improve selectivity and to decrease the side effects on healthy tissues in cancer treatment. Iron oxide (mainly magnetite, $Fe_3O_4$), which presents a low toxicity profile and superparamagnetic behavior, has attractive characteristics for this type of application in biological systems. In this article, synthesis and characterization of magnetite (NP-$Fe_3O_4$) and silica-coated magnetite (NP-$Fe_3O_4$/$SiO_2$) nanoparticles, as well as their biocompatibility via cellular toxicity tests in terms of cell viability, are carefully investigated. MCF-7 cells, which are commonly applied as a model in cancer research, are used in order to define prognosis and treatment specifics at a molecular level. In addition, HaCaT cells (immortalized human keratinocytes) are tested, as they are normal, healthy cells that have been used extensively to study biocompatibility. The results provide insight into the applicability of these magnetic nanoparticles as a drug carrier system. The cytotoxicity of nanoparticles in breast adenocarcinoma (MCF-7) and HaCat cells was evaluated, and both nanoparticles, NP-$Fe_3O_4$/$SiO_2$ and NP-$Fe_3O_4$, show high cell viability (non-cytotoxicity). After loading the anti-tumor drug doxorubicin (Dox) on NP-$Fe_3O_4$/Dox and NP-$Fe_3O_4$/$SiO_2$/Dox, the cytotoxicity against MCF-7 cells increases in a dose-dependent and time-dependent manner at concentrations of 5 and 10 µg/mL. HaCat cells also show a decrease in cell viability; however, cytotoxicity was less than that found in the cancer cell line. This study shows the biocompatibility of NP-$Fe_3O_4$/$SiO_2$ and NP-$Fe_3O_4$, highlighting the importance of silica coating on magnetic nanoparticles and reinforcing the possibility of their use as a drug carrier system against breast adenocarcinoma cells (MCF-7).

**Keywords:** superparamagnetic iron oxide nanoparticles; cancer; biocompatibility; doxorubicin

## 1. Introduction

In 2020, an estimated 19.3 million new cancer cases and nearly 10 million cancer deaths occurred worldwide. Female breast cancer has overtaken lung cancer as the most commonly diagnosed cancer, with an estimated 2.3 million new cases, 11.7% of the total, and the fourth leading cause of cancer-related death [1]. Cell culture lines can be applied in several approaches in research, particularly as *in vitro* models in cancer research, and are useful at the molecular level in order to define the prognosis and specific treatment. In the case of breast cancer, MCF-7 cells are commonly used as a suitable human breast cell line model, including for the development of anticancer drugs [2–4]. Recent cancer treatment studies have been trying to find the best and least risky way to replace the old methods. The main problem with currently used cancer treatments is their adverse side effects on healthy tissue. Nanotechnology, particularly magnetic nanoparticle-based applications, can help to overcome these problems. Site-specific drug administration can ensure safe

and reproducible treatment of diseases in defined, localized positions and can also prevent overdose and corresponding unwanted side effects [5].

Interest in magnetic nanoparticles (MNPs) has been growing continuously due to their unique physico–chemical characteristics, including enhanced magnetic properties, chemical stability, low toxicity and biodegradability [6–8]. In general, their magnetic behavior depends on particle size, particle shape and crystalline structure [9]. In fact, they offer several possibilities of use in the biomedical field, and due to their well-known superparamagnetic behavior, these nanoparticles can be manipulated by external magnetic fields and, consequently, be precisely deposited in a desired location [10], for example, for hyperthermia treatment [11,12] and cancer therapy [13–15]. In addition, they are seen as the next generation of specific drug delivery systems [5,6,14], precisely because under the action of appropriate magnetic fields, these MNPs can be conducted to a specific target [16–18]. Generally, they can be divided into metal oxides, pure metals and magnetic nanocomposites. Among these, magnetic nanoparticles such as $Fe_3O_4$, $Fe_2O_3$, $Fe_xO_y$ and other iron oxides have been exploited as "magnetic carriers" for drug delivery [8,19]. It is known that one of the limitations in delivering drugs is inability to transport the drug directly to the center of the disease or tumor site. Nevertheless, by using MNPs, this issue can be overcome, which would reduce systemic toxicity and side effects [16,18]. In particular, magnetite nanoparticles ($Fe_3O_4$), which show low toxicity profiles and superparamagnetic behavior, have attractive characteristics for this type of application in biological systems [20,21].

It is important to note that nanomaterials are small enough to move around the body without affecting its normal functions and have access to places inaccessible to other materials. However, cells can also react in the presence of nanomaterials, and these reactions can produce changes in cells that lead to cell growth or cell death [22]. Although some MNPs have been approved by the FDA (U. S. Food and Drug Administration), they should not be administered in the body without control. Several toxicity studies have been conducted on MNPs with or without a surface coating and with different coatings [23,24]. Some results have indicated the toxic potential of these MNPs, mainly when used for drug delivery or magnetic resonance imaging [25,26]. MNP toxicity can be mitigated by covering them with different materials, including inorganic and organic coatings [27]. Among these materials, silica may offer significant advantages as a coating due to its inertness, agglomeration prevention, high biocompatibility and improvement of MNP stability [28–30]. In fact, the surface of silica ends with silanols (-SiOH), which allows easy functionalization with different groups (such as amines and carboxylic groups) and can interact with different molecules such as drugs and enzymes that can be used in the biomedical area [31,32].

The application of MNPs as therapeutic carriers can help overcome a number of disadvantages of traditional applications, such as limited effectiveness, poor distribution in the body and low selectivity [5,16]. MNPs offer an attractive means to remotely target therapeutic agents specifically to a disease site, reducing dosage and deleterious side effects associated with non-specific drug absorption by healthy tissues. After magnetic guidance, drug release can be initiated by external stimuli, such as a limited rise in temperature, or by internal stimuli, such as a change in pH [10,33–35]. These approaches are highly promising for cancer therapy because they may enhance the efficacy and decrease the cytotoxicity of drugs that are usually applied in cancer treatment, such as doxorubicin (Dox). Dox is considered one of the most effective chemotherapeutics and is currently approved by the FDA for a variety of cancers [36]. However, its use is hampered by relatively low selectivity for cancer cells and severe side effects due to uptake by healthy cells and tissues [37]. Therefore, targeted drug delivery systems are naturally preferred to increase the functional efficiency of transportation to specific tissues and to reduce potential side effects [16].

Some hierarchical carriers utilizing, for example, mesoporous silica nanocarriers (MSNs) co-impregnated with metallic copper in the silica framework (Cu-MSNs) or Zn-co-impregnated mesoporous siliceous frameworks (Zn-MSNs), which facilitates coordinated interactions to immobilize Dox for its pH-sensitive release, have been used for cancer

cell treatments [38,39]. Several approaches related to iron oxide coated with different materials and loaded with Dox have been analyzed in order to study these materials as drug carriers for cancer therapy [11,32,40]. The amount of Dox loaded in the MNPs is highly dependent of coating nature; i.e., MNPs coated with oxalic acid presented a high loading capacity of 867 µg of Dox per mg of MNPs [41]. In another case, MNPs were stabilized by carboxymethylcellulose sodium salt and presented a loading capacity of approximately 0.1 µg of Dox per mg of MNPs [42]. In both cases, the materials were able to decrease the cell viability of cancerous cells (lung adenocarcinoma, A549 cells and breast cancer cells, respectively) and were time- and dose-dependent. In addition, in the second example, the conjugation of Dox with MNPs promoted less cytotoxicity in healthy breast cells. D. Nieciecka et al. analyzed iron oxide/holmium-based MNPs coated with citric acid and loaded with two chemeotherapeutics (Dox and epirubicin). Again, the cytotoxicity of Dox-loaded MNPs against SKOV-3 cancer cells was time- and dose-dependent (loading capacity of 0.1 µg of Dox per mg of MNPs) [43].

In this study, synthesis and characterization of magnetite nanoparticles and silica-coated magenetite nanoparticles, as well as biocompatibility assays in terms of cell viability and morphology changes by in vitro cellular toxicity tests (HaCat cells used as healthy cells and MCF-7 as breast cancer cells) are carefully conducted. The strategy to enhance the toxicity of Dox on MCF-7 cancer cells by functionalization of these magnetic nanoparticles by electrostatic interactions is proposed. In addition, the advantages of silica as coating for the MNPs are pointed out, and insights into the applicability of these nanoparticles as a drug carrier system are presented and discussed.

## 2. Materials and Methods

### 2.1. Nanoparticle Synthesis and Characterization

#### 2.1.1. Materials

Ferrous chloride tetrahydrate ($FeCl_2.4H_2O$), ferric chloride hexahydrate ($FeCl_3.6H_2O$), tetraethyl orthosilicate (TEOS) and Igepal CO-520 were purchased from Sigma-Aldrich. Ethanol ($C_2H_5OH$), ammonium hydroxide solution ($NH_3$ aq. 28%) and oleic acid ($C_{18}H_{34}O_2$) were purchased from Labsynth. In all syntheses, the aqueous solution was deoxygenated by argon bubbling for 15 min before use.

#### 2.1.2. Magnetic Nanoparticle Synthesis

Magnetic nanoparticles ($NP-Fe_3O_4$) were synthesized by the chemical co-precipitation method [44,45]. Initially, a solution of iron chlorides was prepared using 6 mmol $Fe^{3+}$ and 3 mmol $Fe^{2+}$ in 30 mL of deionized and degassed water. Under stirring and an argon atmosphere, the solution was heated to 80 °C, and then 15 mL of $NH_3$ aq. (28%) was added to the reactional system, instantly forming a black precipitate. The suspension was continuously stirred for 1 h. Finally, the system was cooled to room temperature, and the nanoparticles were magnetically separated. The supernatant was removed, and the nanoparticles were washed several times with deionized water. Oleic acid-coated magnetite nanoparticles ($NP-Fe_3O_4/OA$), used as precursor for coating magnetite nanoparticles, were prepared using the same method as above. In this synthesis, 5 min after the addition of $NH_3$ aq., 3 mmol of OA was added to the medium, and the mix was continuously stirred for 1 h. Posteriorly, the separation process and washing were similar to the process already described for the preparation of $NP-Fe_3O_4$.

#### 2.1.3. Silica-Coated Iron Oxide Nanoparticle Synthesis

The $NP-Fe_3O_4/OA$ nanoparticles were coated with silica using the assisted-microemulsion route of reverse micelles based on the method described by Souza et al. [46]. Firstly, 4 mL of the surfactant Igepal CO-520 was added to 75 mL of cyclohexane. This mixture was stirred for 5 min, and then 7.5 mL of $NP-Fe_3O_4/OA$, dispersed in cyclohexane (10 mg/mL), was added. The suspension was constantly stirred for 30 min, and then 750 µL of TEOS was added slowly under stirring and kept for 30 minutes. Finally, 650 µL of $NH_3$ aq. 28% was added to the

mixture, and the reactional medium was continuously stirred for 24 h at room temperature. Posteriorly, the product of this reaction, silica-coated magnetite nanoparticles ($NP-Fe_3O_4/SiO_2$), was washed several times with ethanol and magnetically separated.

### 2.1.4. *X-ray Diffraction*

The structure and composition of the magnetite nanoparticles were studied by X-ray diffraction (XRD) using a Shimadzu XRD 7000 with $Cu$–$K_\alpha$ radiation equipped with a counter monochromator and at a scattering angle of 2θ from 10° to 70°.

### 2.1.5. *FTIR Spectroscopy*

Infrared spectra were accurately recorded with an FTIR. Powder samples were ground and pressed into pellets. FTIR spectra in the 4000–400 $cm^{-1}$ typical range were acquired by a gradual accumulation of 128 scans with an appropriate resolution of 4 $cm^{-1}$.

### 2.1.6. *Zeta Potential Determination*

Zeta potential ($\zeta$) measurements were performed to investigate the surface charge of magnetic nanoparticles in aqueous medium. The processed nanoparticles were evaluated by a Litesizer 500 analyzer (Anton Paar), and the $\zeta$ values were calculated using Kalliope software version 2.10.6. Standard measurements were carefully performed by dispersing diluted samples of MNPs in distilled water at a concentration of 0.01 mg/L at room temperature.

### 2.1.7. *Vibrating-Sample Magnetometer (VSM)*

The magnetic curves (hysteresis loops) were accurately determined at room temperature using a custom vibrating-sample magnetometer under applied magnetic fields up to 15 kOe [47].

### 2.1.8. *Doxorubicin Loading*

Loading of the water-soluble anticancer drug doxorubicin (Dox) on the nanoparticle surfaces, $NP-Fe_3O_4$ and $NP-Fe_3O_4/SiO_2$, was done by mixing 5 mg of each nanoparticle with 5 mL of Dox solution (0.1 mg/mL in PBS, pH 7.4). This protocol was adapted from the protocol used by S. Kayal et al. [48]. The mixtures of nanoparticles and Dox were stirred for 24 h at room temperature. The magnetic nanoparticles were removed by centrifugation at 12,000 rpm for 30 min, and the supernatant was used to measure the efficiency rate of drug loading. The loading efficiency (*LE*) was determined by UV–vis spectroscopy using a T90 spectrophotometer from PG Instruments Ltd, taking into account the absorbance due to the presence of Dox in the solutions at a wavelength of 480 nm. Drug-loading efficiency was calculated by Equation (1).

$$LE\ (\%)\ =\ \frac{total\ of\ DOX\ (\mu g)\ -\ DOX\ in\ the\ supernadant\ (\mu g)}{total\ of\ DOX\ (\mu g)} \times 100 \tag{1}$$

### 2.2. *Cell Lines*

Cells from a breast adenocarcinoma cell line (MCF-7 - ATCC HTB-22) and human immortalized keratinocytes (HaCat - Cell Lines Service (CLS), 300,493), were cultured separately in 25 $cm^2$ culture flasks containing Dulbecco's Modified Eagle Medium (DMEM, Gibco) complete culture medium supplemented with 10% fetal bovine serum (FBS) and 1mL/L of antibiotic/antimycotic solution and kept in an incubator at 37 °C in a humidified atmosphere with 5% $CO_2$.

### 2.3. *Cytotoxicity Assay–MTT*

The cytotoxic potential of the nanoparticles was determined by the MTT (3- [4,5-[UW1] dimethylthiazol-2-yl] -2,5 diphenyl tetrazolium bromide) assay based on the Mosmann protocol [49]. MCF-7 and HaCaT cells were seeded, separately, at a density of $2.5 \times 10^4$ cells per mL in 96-well plates under standard cell culture conditions in a $CO_2$ incubator at 37 °C

for 24 h. Posteriorly, different concentrations of NP-Fe$_3$O$_4$ and NP-Fe$_3$O$_4$/SiO$_2$ (5, 10, 25 μg/mL) were added, as well as Dox (positive control) and nanoparticles loaded with Dox, NP-Fe$_3$O$_4$/Dox and NP-Fe$_3$O$_4$/SiO$_2$/Dox (5 and 10 μg/mL). For the control, only 10% FBS DMEM culture medium was added to the cells. The plates were maintained in an oven in a humidified atmosphere with 5% CO$_2$ at 37 °C for 24 and 48 h. After each exposure time, the treatments contained in the plates were discarded, and a solution containing MTT (0.167 mg/mL) was added. After 4 h of incubation, a solution of dimethylsulfoxide (DMSO) was added to the wells, and the absorbance of each sample was measured using a Labtech Microplate Reader (model: LT-4000) spectrophotometer at 550 nm. Cell viability was determined as the direct ratio of exposed cells' optical density (OD) to the OD of untreated cells. Mean absorbance rates from eight wells were averaged for each effective concentration analyzed. All experiments were performed in triplicate. The differences among the NPs (NP-Fe$_3$O$_4$ and NP-Fe$_3$O$_4$/SiO$_2$), Dox loaded NPs (NP-Fe$_3$O$_4$/Dox and NP-Fe$_3$O$_4$/SiO$_2$/Dox), Dox free and control (untreated cells) were analyzed using one-way ANOVA followed by the Tukey test, in which $p$-values $\leq 0.05$ were considered significant (a = 0.05, $p < 0.05$, n = 3).

### 2.4. Membrane Integrity Analysis by Flow Cytometry

In this analysis, propidium iodide (PI) was used as a probe that binds to DNA in cells with a non-intact membrane, allowing identification of cell membrane integrity. For the assay, MCF-7 cells were grown in six-well culture plates at a density of $5 \times 10^5$ cells/mL. The cells were exposed for 24h to different NP groups (NP-Fe$_3$O$_4$, NP-Fe$_3$O$_4$/SiO$_2$, NP-Fe$_3$O$_4$-/Dox and NP-Fe$_3$O$_4$/SiO$_2$/Dox), as well as Dox free, at two concentrations: 5 and 10 μg/mL. Untreated cells were used as control. Then, the cells were harvested, centrifuged and re-suspended in phosphate-buffered saline (PBS) and incubated with 5 μL of PI for 10 min, and the integrity of the cell membrane was checked. Data acquisition and analysis were performed using a FACSCalibur flow cytometer equipped with CellQuest software. A total of 5000 events were acquired.

### 2.5. Scanning Electron Microscopy (SEM)

Scanning electron microscopy (SEM) was used to assess morphological changes to the cells after contact with the NPs [37]. MCF-7 cells were seeded in 24-well plates at a density of $2.5 \times 10^4$ cells/mL. The culture cells were exposed for 24h to different NP groups (NP-Fe$_3$O$_4$, NP-Fe$_3$O$_4$/SiO$_2$, NP-Fe$_3$O$_4$/Dox and NP-Fe$_3$O$_4$/SiO$_2$/Dox) and Dox free at concentration of 10 μg/mL. As control was used untreated cells. Posteriorly, the wells were washed once with PBS buffer and fixed by 2.5% glutaraldehyde in 0.1 M phosphate buffer (pH 7.3) for 90 minutes at 8 °C. They were then dehydrated in a gradual series of ethanol (20–100%). After dehydration, they were submitted to the critical point, metalized, and examined under a FEI Quanta 250 scanning electron microscope.

### 2.6. Transmission Electron Microscopy (TEM)

The nanoparticle suspensions were prepared in distilled water, exposed to an ultrasonic bath for 10 min, placed on a carbon grid, and then dried using a suitable solvent at room temperature. The morphology, particle size and accurate characterization of the nanoparticles produced in this study were efficiently performed using a JEOL JEM 1400 microscope. The specific dimensions of a representative number of observed particles (at least 250) were obtained from different TEM images via ImageJ software [50].

The ultrastructural changes to the cells, after the contact with the nanoparticles, were analyzed under the same microscope. MCF-7 cells were seeded in a 6-well plate at a density of $2.5 \times 10^5$ cells/mL and were kept in an oven at 37 °C with 5% CO$_2$ for 24 h. For comparison, untreated cells were also analyzed. After washing with PBS, cells were treated with nanoparticles (NP-Fe$_3$O$_4$, NP-Fe$_3$O$_4$/SiO$_2$, NP-Fe$_3$O$_4$/Dox and NP-Fe$_3$O$_4$/SiO$_2$/Dox) and Dox at concentrations of 10 μg/mL and incubated at 37 °C in a humidified atmosphere with 5% CO$_2$ for 24 h. Then, the wells were washed once in PBS buffer, and the cells

were fixed with 2.5% glutaraldehyde in 0.1 M phosphate buffer (pH 7.3) for 90 min at 8 °C. Cells were re-suspended in sodium cacodylate and centrifuged at 3000 rpm for 5 min. At this point, two drops of 2% osmium tetroxide plus potassium ferrocyanide were added. After 1 h, the cells were centrifuged at 10,000 rpm for 3 min and re-suspended in cacodylate buffer, followed by dehydration in increasing concentrations of acetone (30–100%), and the addition of EPON resin. Subsequently, ultra-thin cuts and staining with uranyl acetate and lead citrate were performed, and finally the cells were analyzed by TEM.

### 3. Results and Discussions

#### 3.1. X-ray Diffraction (XRD)

XRD results for uncoated iron oxide, NP-Fe$_3$O$_4$ and the silica-coated iron oxide nanoparticles (NP-Fe$_3$O$_4$/SiO$_2$) are shown in Figure 1a,b. The standard XRD pattern for microscopic magnetite (JCPDS 88-0315) is also shown in Figure 1. As seen, the obtained results are in close agreement with the standard XRD pattern and indicate the formation of an inverse cubic spinel structure. The appearance of well-defined Bragg's peaks indicates that nanoparticles with a high degree of crystallinity were successfully synthesized [51]. In Figure 1b, besides the characteristic peaks of iron oxide, a broadened halo centered at $2\theta$~23° is observed and is related to the characteristic thin silica shell (JCPDS 82-1554) that coats the iron oxide nanoparticles [52]. The mean crystallite sizes for both samples, NP-Fe$_3$O$_4$ and NP-Fe$_3$O$_4$/SiO$_2$, were determined from line-broadening of the most intense diffraction peak ((311) at ($2\theta$~35.5°)) by using the Scherrer's equation [53]. The estimated crystallite sizes are 10.3 ± 0.8 nm and 8.3 ± 0.6 nm for NP-Fe$_3$O$_4$ and NP-Fe$_3$O$_4$/SiO$_2$, respectively. It is important to highlight that NP-Fe$_3$O$_4$ presents a black color, which is one more indication of the predominance of a magnetite structure instead a maghemite structure (red color).

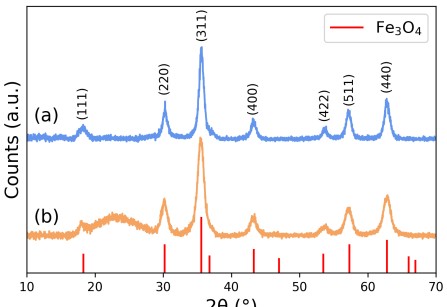 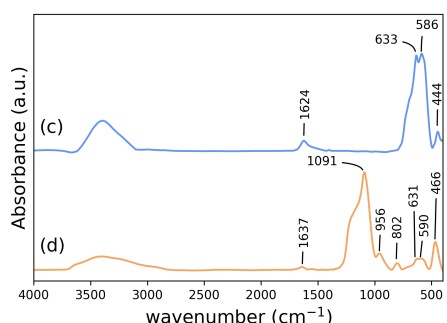

**Figure 1.** XRD patterns for (**a**) NP-Fe$_3$O$_4$ and (**b**) NP-Fe$_3$O$_4$/SiO$_2$ with magnetite pattern from the JCPDS database (JCPDS 88-0315). FTIR spectra for (**c**) NP-Fe$_3$O$_4$ and (**d**) NP-Fe$_3$O$_4$/SiO$_2$ samples.

#### 3.2. FTIR

In order to characterize NP-Fe$_3$O$_4$ and NP-Fe$_3$O$_4$/SiO$_2$ , and to emphasize the presence of silica on the nanoparticle surface (shell), infrared spectra were recorded. The NP-Fe$_3$O$_4$ sample (Figure 1c) has characteristic absorption bands at 633, 586 and 444 cm$^{-1}$ that can be assigned to stretching vibrations of Fe-O bonds at the tetrahedral and octahedral sites, a signature of magnetite nanoparticles. For the silica-coated sample, NP-Fe$_3$O$_4$/SiO$_2$ (Figure 1d), these bands shift to 631, 590 and 466 cm$^{-1}$, respectively. This result suggests the formation of Si–O–Fe bonds on the surface of iron oxide nanoparticles. The high-intensity peak at 466 cm$^{-1}$ also indicates the overlap of the Si–O–Si bending modes. Furthermore, the characteristic absorption bands at 802, 956 and 1091 cm$^{-1}$ are assigned to the symmetric stretching mode of Si–O–Si, symmetric stretching mode of Si–OH, and the bending mode of the Si–O band, respectively [54,55].

#### 3.3. Transmission Electron Microscopy (TEM)

Figure 2 shows the TEM images for uncoated NP-Fe$_3$O$_4$ (Figure 2a) and silica-coated nanoparticles, NP-Fe$_3$O$_4$/SiO$_2$ (Figure 2b). The TEM image for NP-Fe$_3$O$_4$ (Figure 2a) clearly

shows nanoparticles with a nearly spherical shape and an average diameter of ~9.6 nm, as shown in the histogram of size measurements (insert). The size of the nanoparticles described in this study are comparable to that of iron oxide nanoparticles related by S. Ali et al. [56], which used a similar process of synthesis (~9.1 nm). In Figure 2b, the TEM image for the NP-Fe$_3$O$_4$/SiO$_2$ sample reveals a spherical core/shell structure and total average diameter of ~18.1 nm, as shown in the histogram of size measurements (insert). In this case, the core is formed by a magnetite nanoparticle (~8 nm in diameter), and the shell is formed by a ~5 nm thin SiO$_2$ layer.

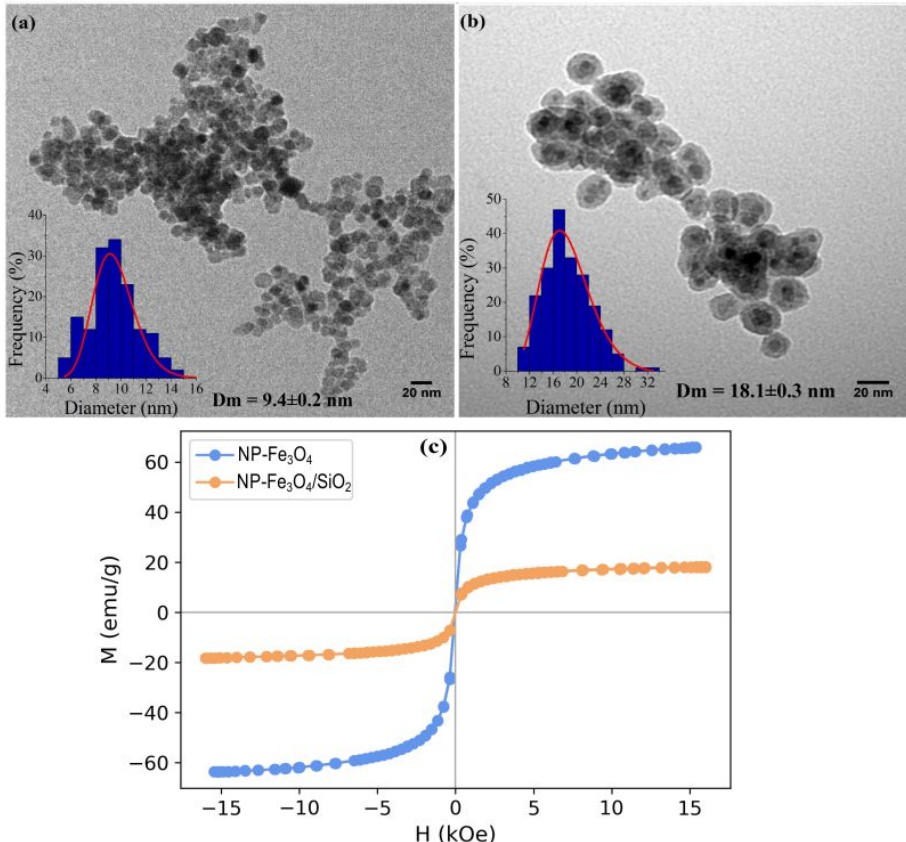

**Figure 2.** TEM images for (**a**) uncoated NP-Fe$_3$O$_4$ and (**b**) silica-coated (NP-Fe$_3$O$_4$/SiO$_2$) magnetite nanoparticles. (**c**) M × H curves at room temperature for NP-Fe$_3$O$_4$ and NP-Fe$_3$O$_4$/SiO$_2$ nanoparticles.

### 3.4. Zeta Potential Analysis

Zeta potential ($\zeta$) measurements were performed in an aqueous dispersion at pH 7. For NP-Fe$_3$O$_4$, the mean superficial potential value was found as $-18.8 \pm 0.8$ mV, while for NP-Fe$_3$O$_4$/SiO$_2$, the mean superficial potential was $-28.1 \pm 0.5$ mV. The decrease in the $\zeta$ value indicates an increase in coulombic repulsion caused by the presence of the silica layer on the surface of the magnetite nanoparticles (presence of hydroxyl groups), which improves the stability of the nanoparticles in the aqueous suspension [57].

### 3.5. Vibrating Sample Magnometometer

Magnetic hysteresis curves for NP-Fe$_3$O$_4$ and NP-Fe$_3$O$_4$/SiO$_2$ nanoparticles are shown in Figure 2c. As can be seen, the saturation magnetization ($M_s$) obtained for the synthesized magnetite is lower than that of multidomain bulk magnetite (~90 emu/g) [58]. Remarkably, different particle sizes, surface areas, crystal defects and chemical compositions can properly explain possible deviations in the saturation magnetization. In fact, $M_s$ values reached 63.8 and 18.2 emu/g, respectively, for NP-Fe$_3$O$_4$ and NP-Fe$_3$O$_4$/SiO$_2$ samples. The literature provides saturation magnetization values for similar SiO$_2$-coated magnetite

nanoparticles close to the ones measured in this study [52,59]. The insert in Figure 2c shows an enlarged view of low field of hysteresis curves for both samples. As can be seen, the analyzed nanoparticles present coercivity close to zero and have high remanence, indicating superparamagnetic behavior at room temperature. In most cases, an inorganic layer on the surface of nanoparticles can undoubtedly contribute to an apparent decrease in $M_s$ values since the magnetic signal is proportional to the mass in standard VSM tests [60]. $M_s$ values can properly be corrected by considering only the mass of the magnetic core. The mass of each component of the core/shell structure can be estimated by taking into account the density of the materials. In the case of the $SiO_2$ shell, a density of 1.87 $g/cm^3$ is assumed [61,62], with 5.10 $g/cm^3$ for the magnetic core (magnetite) [63]. Using the mean diameter obtained by TEM images for the core/shell nanoparticles (NP-$Fe_3O_4$/$SiO_2$), the contribution of each component for $Ms$ is estimated as 78.6% for the silica shell and 21.4% for the magnetite core. Therefore, the $M_s$ can be properly corrected and reaches 85.07 emu/g, i.e., increasing by ~33.2% in comparison with NP-$Fe_3O_4$ nanoparticles. In fact, NP-$Fe_3O_4$ can typically undergo an oxidative process, and thus, their outer layer (surface) can invariably lose its magnetic properties. However, in NP-$Fe_3O_4$/SiO2 samples, the $SiO_2$ on the particles' surface acts as a protective layer, preventing oxidation.

### 3.6. Drug-Loading Efficiency

To further explore the application of the obtained nanoparticles as anticancer drug carriers, the loading efficiency of Dox was determined. Many works have reported the strong affinity of Dox for various negatively charged groups, such as carboxylates, oleates and phospholipids, mainly due to electrostatic interaction [64–67]. In this study, the conjugation of Dox onto NP- $Fe_3O_4$ and NP-$Fe_3O_4$/$SiO_2$ surfaces was carried out following the procedure used by S. Kayal et al. with some modifications and was analyzed using UV–vis spectroscopy [48]. It is believed that the driving force for assembly was primarily the electrostatic interaction between Dox and groups on the surfaces of the nanoparticles. The electrostatic interaction between the two distinct entities can be explained as a direct result of the amine group portions of Dox and the negative charge of the groups on the nanoparticles' surface. This negative charge was evidenced by the negative zeta potential of the surface. The loading efficiency (*LE*) was determined for both samples and indicated a Dox loading of 85% and 69%, respectively, for NP-$Fe_3O_4$ and NP-$Fe_3O_4$/$SiO_2$. These percentages effectively represent 85 μg of Dox per mg of NP-$Fe_3O_4$, and 69 μg of Dox per mg of NP-$Fe_3O_4$/$SiO_2$. The amount of Dox loaded onto NP- $Fe_3O_4$ and NP-$Fe_3O_4$/$SiO_2$ surfaces is slightly higher than the values achieved by S. Kayal et al. (maximum of 58 μg/mg) [48]. The strategy to use PBS at pH 7.4 as solvent also contributed to stabilize Dox and improved the loading on the nanoparticles' surface. This fact was equally verified in other studies that analyzed the adsorption of Dox on silica particles [28,68], as well as in the case of Dox-loading onto polymer-coated iron oxide [35]. In Lungu et al. [42], the efficiency of DOX loading was also analyzed by laser-induced fluorescence. A concentration of 0.8 μg/mL in a 1:10 dilution of DOX-loaded $\gamma$-$Fe_2O_3$ suspension was found, corresponding to a loading efficiency of 10%. Kovrigina et al. proposed two possible drug-loading conditions at a basic pH [41]. The first provides excellent capacity, up to 1757 μg DOX/1 mg for magnetic nanoparticles in oleic acid, with a 24% DOX loading efficiency. Up to 870 μg DOX/1 mg for magnetic nanoparticles in oleic acid with ~90% DOX loading efficiency was found for the second approach.

### 3.7. Cell Viability Assay

The cytotoxicities of NP-$Fe_3O_4$ and NP-$Fe_3O_4$/$SiO_2$ formulations toward human breast cancer (MCF-7) and normal keratinocyte (HaCat) cells were examined by using the thiazolyl blue tetrazolium bromide (MTT) cell viability assay, primarily to demonstrate the potential of these MNPs as biocompatible drug carriers. The cell viability assays on MCF-7 (Figure 3a,b) and HaCat cells (Figure 3c,d) in the presence of NP-$Fe_3O_4$ and NP-$Fe_3O_4$/$SiO_2$ unequivocally show no cytotoxicity in the tested cells in either analyzed period

(24 h and 48 h). In fact, cell viability was higher than 70% (as established by the ISO 10993-5 standard) in all analyzed conditions. The 70% limit is indicated by the red dashed lines in Figure 3. In addition, it was possible to verify the benefits of silica coating on NP-Fe$_3$O$_4$ in the tests with healthy cells (HaCat), as after 24 h, the cell viability for NP-Fe$_3$O$_4$ was lower (70%) than for NP-Fe$_3$O$_4$/SiO$_2$. In all cases, cell viability reached a minimum of 85% when exposed to silica-coated iron oxide (NP-Fe$_3$O$_4$/SiO$_2$). E. Helal-Neto et al. analyzed exposure of silica-coated magnetic nanoparticles to different cancerous cell lines and healthy cells, and they verified that these nanoparticles have no cytotoxicity at the concentrations analyzed [69], which corroborates with the results presented in our work. Other studies with mesopourous silica nanoparticles similarly reported a decrease in cell viability by 20% in MCF-7 cell lines and HeLa cells at a concentration of 10 µg/mL [70,71], which supports our observations and confirms the biocompatibility of the analyzed nanoparticles.

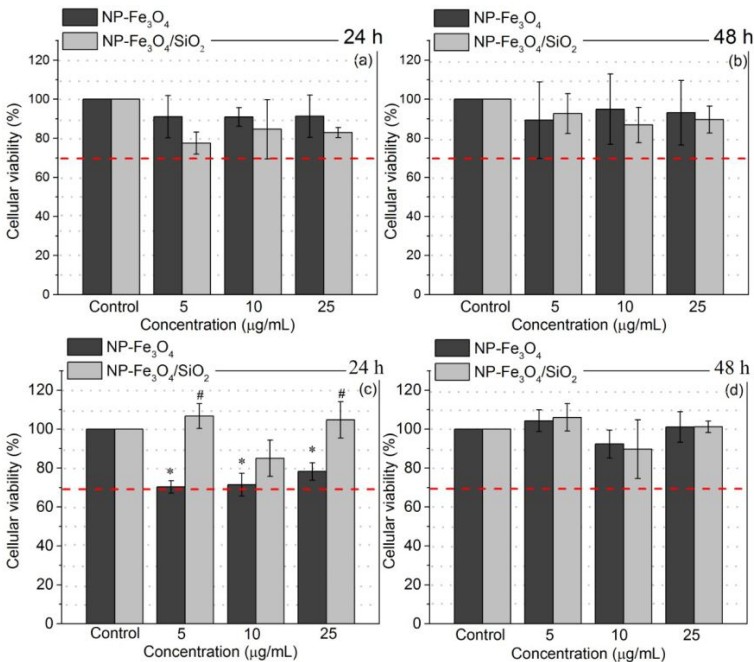

**Figure 3.** Cell viability assays for MCF-7 cells treated with NP- Fe$_3$O$_4$ and NP-Fe$_3$O$_4$/SiO$_2$ at different NP concentrations (5 to 25 µg/mL) for (**a**) 24 h and (**b**) 48 h. Control group: untreated MCF-7 cells. Cell viability assays for HaCaT cells treated with NP- Fe$_3$O$_4$ and NP-Fe$_3$O$_4$/SiO$_2$ at different NP concentrations (5 to 25 µg/mL) for (**c**) 24 and (**d**) 48 h. Data are expressed as percentage of control. * Statistically significant difference in relation to control ($p < 0.05$); # Statistically different in relation to 5 µg/mL NP-Fe$_3$O$_4$ and statistically different compared to 25 µg/mL NP-Fe$_3$O$_4$ ($p < 0.05$). Red dashed line threshold shows the limit percentage of cell viability (70%, ISO 10993-5).

Figure 4 shows similar cytotoxicity assays for the presented MNPs loaded with Dox. It is worth highlighting that different drug-loaded nanoparticle formulations (5 µg/mL and 10 µg/mL) were found to be highly toxic to MCF-7 breast cancer cells, with their lowest cytotoxicity found in normal keratinocyte cells (HaCat). The MNPs loaded with Dox (Figure 4) decreased cell viability compared to the control at all exposure times, specially at the concentration of 10 µg/mL. The Dox effective concentrations at 5 and 10 µg/mL nanoparticle formulations are, respectively, 0.392 and 0.783 µg for NP-Fe$_3$O$_4$/Dox and 0.323 and 0.645 µg for NP-Fe$_3$O$_4$/SiO$_2$/Dox. In this sense, the effective concentration of Dox in the MNPs is much lower than that of free Dox used in the control sample. These results indicate that Dox-loaded nanoparticles are very effective in decreasing MCF-7 cancer cell viability. It can also be understood that, considering nanoparticle internalization, NP-Fe$_3$O$_4$/Dox and NP-Fe$_3$O$_4$/SiO$_2$/Dox more effectively release Dox inside the MCF-7 cancer cells, and consequently enhance cytotoxicity through synergistic anti-proliferative activities in the cancer cell line. In addition, time dependence for drug release inside the

cells for loaded MNPs was noted. After 48 h of treatment, cell viability decreased to around 10% for the 10 µg/mL sample of both Dox-loaded MNPs. Furthermore, coating with silica seems to improve the internalization of nanoparticles in the cells. Compared to data in the literature in which nanoparticles were coated with other materials then loaded with Dox (in quantities equivalent to our case), such as (Trimethoxysilylpropyl)-ethylenediamine triacetic acid (EDT) (analysis using glioblastoma cells) [40] and PVCL-co-PAA copolymer (analysis using MCF-7 cells) [35], NP-$Fe_3O_4$/$SiO_2$/Dox turned out to be more effective in decreasing cancer cell viability. These results emphasize the enhanced cytotoxicity to cancer cells of the processed MNPs loaded with Dox in comparison to free drug; further, we had the least cytoxicity in normal cells, suggesting huge potential for NP-$Fe_3O_4$/Dox and NP-$Fe_3O_4$/$SiO_2$/Dox as selective and efficient drug delivery carriers. Additionally, these results corroborate other cases in vitro release of Dox-loaded nanoparticles found in the literature, in which the release is mainly dependent on the pH of the medium, and it is well-known that the pH of cancerous cells is lower that of healthy cells [32,40,72,73].

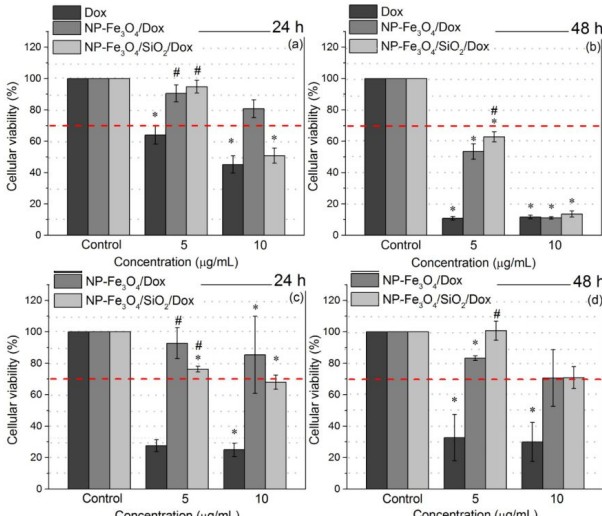

**Figure 4.** Cell viability assay for MCF-7 cells treated with Dox, NP-$Fe_3O_4$/Dox and NP-$Fe_3O_4$/$SiO_2$/Dox at concentrations of 5 and 10 µg/mL for (**a**) 24 h and (**b**) 48 h. Control group: untreated MCF-7 cells. Cell viability assays for HaCat cells treated with Dox, NP-$Fe_3O_4$/Dox and NP-$Fe_3O_4$/$SiO_2$/Dox at concentrations of 5 and 10 µg/mL for (**c**) 24 h and (**d**) 48 h. Control group: untreated HaCat cells. Data are expressed as percentage of control group. \* $p < 0.05$, significantly different compared to control; # $p < 0.05$, significantly different compared to 5 µg/mL doxorubicin; $p < 0.05$, significantly different compared to 10 µg/mL doxorubicin. Red dashed line threshold shows the limit percentage of cell viability (70%, ISO 10993-5).

### 3.7.1. Plasma Membrane Integrity

In order to analyze the plasma membrane integrity of MCF-7 cells, flow cytometry tests were performed. These tests measured propidium iodide uptake by cells subjected to 24 h of treatment with different concentrations of the nanoparticle systems synthesized in this study. Propidium iodide uptake only occurs in cells that have suffered some type of membrane damage. Figure 5 shows the number of positive and negative PI cells, where negative PI indicates cells with viable or intact membranes. As can be seen, there was an increase in PI-positive cells after contact with Dox, NP-$Fe_3O_4$/Dox and NP-$Fe_3O_4$/$SiO_2$/Dox, suggesting that damage to the cell membrane occurred. It is possible to verify both the effect of Dox-loaded nanoparticle concentration and the higher number of cells damaged when exposed to NP-$Fe_3O_4$/$SiO_2$/Dox. However, the levels of PI positive cells was similar to that of the control sample for MCF-7 cells exposed to NP-$Fe_3O_4$ and NP-$Fe_3O_4$/$SiO_2$. These results suggest that the processed MNPs with no drug loaded did not damage the cell membrane, corroborating the cell viability results.

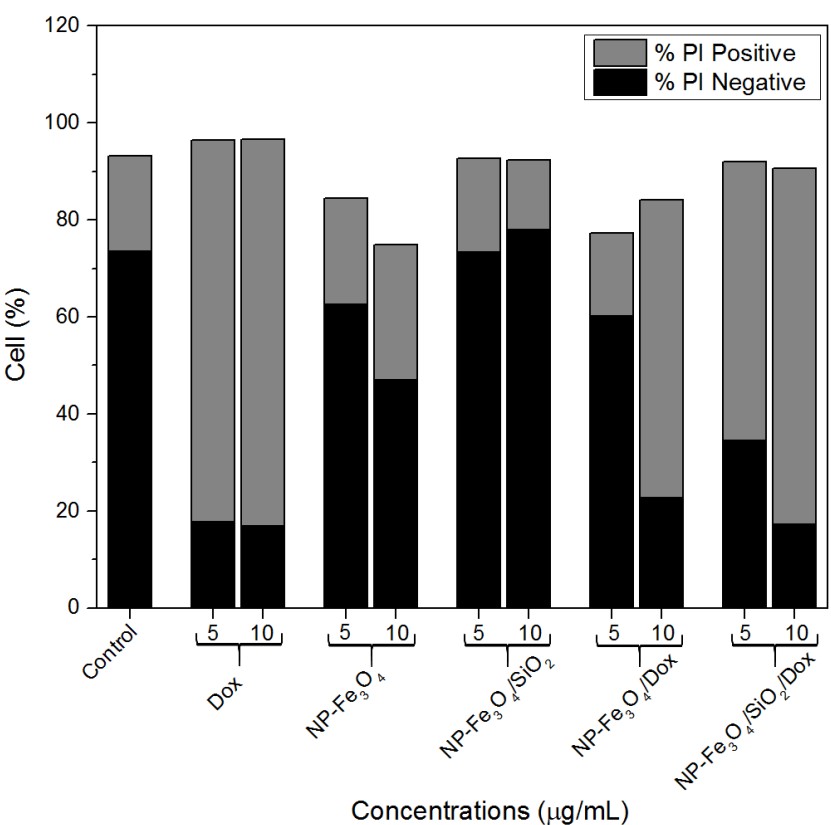

**Figure 5.** Membrane integrity tests of MCF-7 cells subjected to different treatments using 5 and 10 μg/mL for 24 h. From left to right: control; doxorubicin (Dox); NP-Fe$_3$O$_4$; NP-Fe$_3$O$_4$/SiO$_2$); NP-Fe$_3$O$_4$/Dox; and NP-Fe$_3$O$_4$/SiO$_2$/Dox.

3.7.2. Scanning Electron Microscopy—Cell Morphology

To further study the morphology of MCF-7 cells after 24 h of treatment with Dox, NP-Fe$_3$O$_4$, NP-Fe$_3$O$_4$/SiO$_2$, NP-Fe$_3$O$_4$/Dox and NP-Fe$_3$O$_4$/SiO$_2$/Dox, SEM analyses were carried out to identify the cellular surface structure and shape. Figure 6A,B show that the distinct SEM images for the control group (MCF-7 cells) depict healthy cells firmly adhered to the substrate with an abundance of short microvilli distributed across the cell surface and with cells well-attached in all directions. Figure 6C,D show the MCF-7 cells under Dox treatment. It is seen that the cell membrane ruptured, damaging the cell. Regarding the cells treated with NP-Fe$_3$O$_4$ and NP-Fe$_3$O$_4$/SiO$_2$, SEM images (Figure 6E,F show that most cells had extensions of their body, presenting a morphology similar to the untreated control even after being in contact with 10 μg/mL of both studied MNPs. In addition, MNPs can be seen attached to the cell membrane. Parameters such as outer membrane surface structure, adhesion and overall shape do not appear to have been altered by the nanoparticles, even drug-loaded nanoparticles (Figure 6G,H). Despite NPs appearing to be attached to the cell membrane, wrapping of the cell membrane around particles is evident (Figure 6C–F), showing that the processed and studied MNPs are cell-associated and interact with the plasma membrane, and that some of them were probably internalized into the cells.

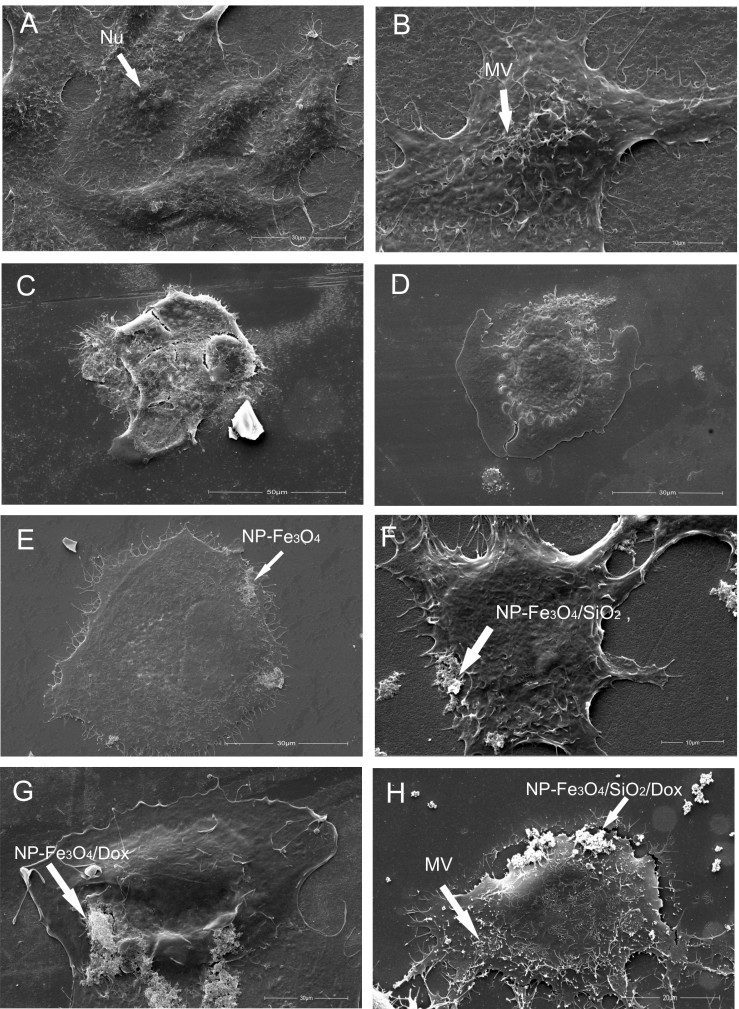

**Figure 6.** Scanning electron microscopy images of MCF-7 cells that stayed in contact with different analyzed systems for 24 h. (**A**,**B**) Control, bar: 15 μm, 10 μm, respectively; (**C**,**D**) Dox 10 μg/mL, bar: 25 μm, 15 μm, respectively; (**E**) NP-Fe$_3$O$_4$ 10 μg/mL, bar: 15 μm; (**F**) NP-Fe$_3$O$_4$/SiO$_2$ 10 μg/mL, bar: 10 μm; (**G**) NP-Fe$_3$O$_4$/Dox 10 μg/mL, bar: 10 μm; and (**H**) NP-Fe$_3$O$_4$/SiO$_2$/Dox 10 μg/mL, bar: 10 μm; MV, microvilli; Nu, nucleolus.

### 3.7.3. Transmission Electron Microscopy—NP Distribution Inside Cells

Figure 7 shows images of the studied cells: (A) control MCF-7 cells and (B) 10 μg/mL Dox-treated MCF-7 cells (24 h). Figure 7 also enables us to understand the internalization process and track the fate of the studied MCF-7 cells in contact with (C) NP-Fe$_3$O$_4$, (D) NP-Fe$_3$O$_4$/Dox, (E) NP-Fe$_3$O$_4$/SiO$_2$ and (F) NP-Fe$_3$O$_4$/SiO$_2$/Dox (10 μg/mL/24 h). Figure 7C,D show the spatial distribution of nanoparticles (dark circles) of 9 nm mean diameter. Figure 7E,F also show internalized nanoparticles with 18 nm diameter within intracellular vesicles, indicating the presence of an active uptake mechanism such as endocytosis. The internalization of magnetic nanoparticles loaded or unloaded with Dox into cancerous cells was also verified by R.C. Popescu et al. [14] when they exposed HeLa cells to PEG-coated iron oxide nanoparticles loaded and unloaded with Dox. Studies have shown that particle internalization and subsequent routing are highly dependent on particle size, shape, composition and surface properties, and parameters such as cell type, protein expression level and cell-cycle phase. As shown here, the size-dependent interaction of different particles with the cell membrane is likely related to the encapsulation process of the incipient membrane. Despite significant efforts in this area, linking specific cellular responses to particle size remains challenging and unclear [74].

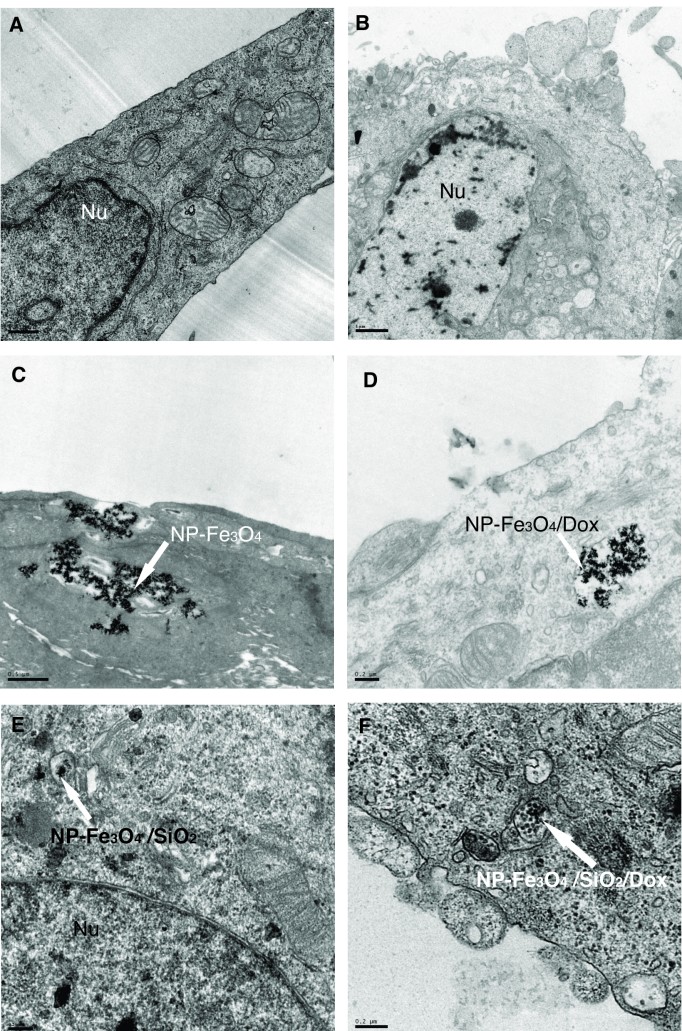

**Figure 7.** TEM images of MCF-7 cells that stayed in contact for 24 h with different systems at a concentration of 10 μg/mL: (**A**) control (cells alone); (**B**) Dox; (**C**) NP-Fe$_3$O$_4$; (**D**) NP-Fe$_3$O$_4$/Dox, bar: 10 μm; (**E**) NP-Fe$_3$O$_4$/SiO$_2$; and (**F**) NP-Fe$_3$O$_4$/SiO$_2$/Dox; Nu, nucleus.

Drug-loaded nanoparticles such as NP-Fe$_3$O$_4$/Dox and NP-Fe$_3$O$_4$/SiO$_2$/Dox promise to improve therapeutic efficacy with optimal drug doses and open new avenues for *in vivo* therapy. One of the main advantages of nanotechnology drug delivery systems is increased selectivity, resulting in a decrease of the harmful side effects associated with low specificity that typically limits drug dosages. Furthermore, magnetic guidance is potentially possible in this system using an external magnetic field at the target site after injecting this type of drug-loaded magnetic nanoparticles.

In many cases, treatments with these iron oxide nanoparticles do not generate negative effects on the body, but there is a possibility that cellular overload with these nanoparticles may trigger adverse cellular responses, despite iron oxide nanoparticles not presenting toxicity at concentrations lower than 100 μg/mL [75]. Further, it is important to consider that some cells are more sensitive to treatment with nanoparticles than others, such as demonstrated by Poller et al. [24] and Malvindi et al. [76], where different responses were observed depending on the cell line used. In this case, it is also necessary to tests these nanoparticles using other cell types to better understand their mechanism of action. Magnetic nanoparticle studies are recent, and in spite of all the positive evidence, it is still necessary to evaluate the cytotoxicity of these materials. Finally, our results are relevant in the development of safer magnetic nanoparticles that can be used in several biomedical applications.

## 4. Conclusions

In this work, magnetite nanoparticles were synthesized by chemical co-precipitation. Silica-coated NP-$Fe_3O_4$ was obtained using assisted-microemulsion of reverse micelles. In order to obtain a material for treatment of a specific target, the anticancer drug doxorubicin (Dox) was successfully loaded on NP-$Fe_3O_4$ and NP-$Fe_3O_4$/$SiO_2$ surfaces with high efficiency. This is undoubtedly an essential way to realistically achieve effective damage to cancer cells by ensuring a sufficiently high concentration of nanoparticles in the tumor tissue, regardless of the specific location of the nanoparticles at the cellular level. Regarding the studied treatments, NP-$Fe_3O_4$ and NP-$Fe_3O_4$/$SiO_2$ presented noncytotoxicity to MCF-7 and HaCat cells at the analyzed concentrations. For nanoparticles loaded with Dox, NP-$Fe_3O_4$/Dox and NP-$Fe_3O_4$/$SiO_2$/Dox, treatments of MCF-7 cells showed a cytotoxic profile and higher toxicity toward the tumor cell line. Due to the spatial distribution and concentration of Dox-loaded magnetic nanoparticles on MCF-7 cells, membranes were ruptured, and consequently, cells were damaged. Thus, we can conclude that these nanoparticles can potentially be used to drive the drug to a specific location and thereby lessen the side effects caused by Dox. Although modifications to the formulations of such nanoparticles should be considered in order to further improve their biocompatibility and specificity, NP-$Fe_3O_4$ and NP-$Fe_3O_4$/$SiO_2$ represent building blocks from which a variety of functionalized magnetic nanoparticles can be created. Consequently, this proper establishes cellular uptake and toxicity, adequately providing a baseline from which other surface modifications can be evaluated for potential drug delivery applications for the effective treatment of breast cancer. In fact, these magnetic nanoparticles can be used as site-specific drug delivery vehicles, ensuring safe and reproducible therapy for complex diseases in distinct, localized positions while preventing overdose and corresponding unwanted side effects.

**Author Contributions:** Conceptualization, P.N.d.O., V.E.P.V. and L.F.C.; Methodology, I.V.d.A., G.S.D. and I.A.d.S.; Validation, I.V.d.A., G.S.D. and I.A.d.S.; formal analysis, E.P.H., R.D.B., K.M.E. and V.A.d.O.J.; investigation, E.P.H., R.D.B., K.M.E. and V.A.d.O.J.; writing—original draft preparation, E.P.H., R.D.B., K.M.E., V.A.d.O.J. and V.E.P.V.; writing—review and editing, G.S.D., P.N.d.O. and L.F.C.; supervision, P.N.d.O., V.E.P.V. and L.F.C.; project administration, L.F.C.; funding acquisition, I.A.d.S. and L.F.C. All authors have read and agreed to the published version of the manuscript.

**Funding:** This research was funded by Coordenação de Aperfeicoamento de Pessoal de Nível Superior: Fellowship funding; National Council for Scientific and Technological Development: Project Funding; Fundação Araucária: Project Funding; Financiadora de Estudos e Projetos: Project Funding.

**Acknowledgments:** For their financial support, the authors would like to acknowledge the Brazilian funding agencies CAPES, CNPq and Funda¸cão Araucária. Finally, we thank COMCAP/UEM facility and Finep for the experimental characterizations.

**Conflicts of Interest:** The authors declare no conflict of interest.

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
