# Peer review of "Doxorubicin-Loaded Magnetic Nanoparticles: Enhancement of Doxorubicin’s Effect on Breast Cancer Cells (MCF-7)"

_magnetochemistry, doi:10.3390/magnetochemistry8100114_

Round 1
Reviewer 1 Report
The authors presented the paper "Effects of Magnetic Nanoparticles on Enhance Doxorubicin Treatments in Breast Cancer Cells (MCF-7)". The research is interesting. However, I have some specific comments
1) Much more fresh 2-3 years paper should be presented in the Introduction section.
2) There are many papers about cancer treatment using magnetic nanoparticles and doxorubicin. I think some of the relevant papers have to be cited. For example,
https://www.mdpi.com/search?q=doxorubicine+loaded+iron+oxide+nanoparticles
High Drug Capacity Doxorubicin-Loaded Iron Oxide Nanocomposites for Cancer Therapy
Smart Magnetic Nanocarriers for Multi-Stimuli On-Demand Drug Delivery
Hybrid System for Local Drug Delivery and Magnetic Hyperthermia Based on SPIONs Loaded with Doxorubicin and Epirubicin
Doxorubicin-Conjugated Iron Oxide Nanoparticles Synthesized by Laser Pyrolysis: In Vitro Study on Human Breast Cancer Cells
3) Much more 2020-2022 years paper about magnetite nanoparticles and doxorubicin for cancer treatment should be cited. Moreover, the authors should compare their results with these papers. For example, MTT test in the papers
Doxorubicin-Conjugated Iron Oxide Nanoparticles Synthesized by Laser Pyrolysis: In Vitro Study on Human Breast Cancer Cells
High Drug Capacity Doxorubicin-Loaded Iron Oxide Nanocomposites for Cancer Therapy. In this paper, the authors have synthesized oleic acid coated magnetic nanoparticles with Doxoribicin.
4) Section 2.1.8. Why you have used such drug-loading procedure?
"Dox aqueous solution" What is the pH value, salts? It is important.
As I see in the section 3.6. you have a high loading efficiency, 85% and 69%. It looks like the saturation is nearby. The capacity is only 0.085 mg of Dox per mg of NP-Fe3O4, and 0.069 mg of Dox per mg of 311 NP-Fe3O4/SiO2. According to the paper https://www.mdpi.com/2312-7481/8/5/54/htm It can be obtained much more capacity, which is excellent for cancer treatment.
Of course, I understand that it will be difficult to do the new experiments. It might be useful for future experiments Some references for the loading procedure and some extensive discussion have to be added.
5) In the abstract and paper's title, you have to mention about what type of magnetic nanoparticles the material is going. For example, may be better oleic acid coated magnetite nanoparticle...
6) I have seen zeta potential analysis. Have you measured the sizes of your nanoparticles using dynamic light scattering (DLS)? If yes, the results should be presented in the paper.
7) Cell Viability assay
Why you have not so good results of cell viability (~70%), I mean that you Dox-loaded MNPs are not highly effective? Can you compare the results with other papers and add some discussion?
8) The novelty of the work have to mention in an extensive way in abstract and conclusion part. There are many doxorubicin-loaded MNP and MNP-SiO2 for cancer treatment. The references to these works should be presented in the Results and Discussion part.
Minor comments
line 104 no dot, line 113 two dots, line 102 subscript
NH4OH much more relevant NH3 aq in this solution only 1-2% of NH4+ ions
Why are you present Fig 7 after the Conclusion?
Author Response
Dear Reviewer,
We would like to thank you for the revision on our submitted manuscript ‘‘Doxorubicin loaded magnetic nanoparticles: enhancement of the Doxorubicin effect on breast cancer cell (MCF-7)’’ (magnetochemistry-1869842). The questions were carefully analysed, and the manuscript was modified (improved). Please find below our specific comments.
1) Much more fresh 2-3 years paper should be presented in the Introduction section.
2) There are many papers about cancer treatment using magnetic nanoparticles and doxorubicin. I think some of the relevant papers have to be cited. For example,
https://www.mdpi.com/search?q=doxorubicine+loaded+iron+oxide+nanoparticles
- High Drug Capacity Doxorubicin-Loaded Iron Oxide Nanocomposites for Cancer Therapy
- Smart Magnetic Nanocarriers for Multi-Stimuli On-Demand Drug Delivery
-Hybrid System for Local Drug Delivery and Magnetic Hyperthermia Based on SPIONs Loaded with Doxorubicin and Epirubicin
-Doxorubicin-Conjugated Iron Oxide Nanoparticles Synthesized by Laser Pyrolysis: In Vitro Study on Human Breast Cancer Cells
3) Much more 2020-2022 years paper about magnetite nanoparticles and doxorubicin for cancer treatment should be cited. Moreover, the authors should compare their results with these papers. For example, MTT test in the papers
- Doxorubicin-Conjugated Iron Oxide Nanoparticles Synthesized by Laser Pyrolysis: In Vitro Study on Human Breast Cancer Cells
- High Drug Capacity Doxorubicin-Loaded Iron Oxide Nanocomposites for Cancer Therapy. In this paper, the authors have synthesized oleic acid coated magnetic nanoparticles with Doxoribicin.
Comments to questions 1, 2 and 3:
New references were added to the manuscript and are signalled in the latest version.
The manuscript was improved by the comparison of our results with the literature, and these points are shown in the new version of the work.
In our study we carefully analyse the iron oxide nanoparticles (NP-Fe3O4) and silica coated iron oxide (NP-Fe3O4/SiO2). Fe3O4 coated with oleic acid is an intermediary product used to properly obtain silica coated iron oxide. Taking it in to account, the direct comparison with the paper cited by the reviewer is unpossible (High Drug Capacity Doxorubicin-Loaded Iron Oxide Nanocomposites for Cancer Therapy). We try to explain better why our drug loading and effective delivery are very efficient in the respective sections: 3.6. Drug loading efficiency and 3.7. Cell viability assay. The improvement is highlighted in the current version of the manuscript.
4 - Section 2.1.8. Why have you used such drug-loading procedure?
"Dox aqueous solution" What is the pH value, salts? It is important.
As I see in the section 3.6. you have a high loading efficiency, 85% and 69%. It looks like the saturation is nearby. The capacity is only 0.085 mg of Dox per mg of NP-Fe3O4, and 0.069 mg of Dox per mg of 311 NP-Fe3O4/SiO2. According to the paper: https://www.mdpi.com/2312-7481/8/5/54/htm
It can be obtained much more capacity, which is excellent for cancer treatment. Of course, I understand that it will be difficult to do the new experiments. It might be useful for future experiments. Some references for the loading procedure and some extensive discussion have to be added.
Comments to question 4:
The loading procedure was performed based in what is related in a well cited paper [1] (more than 400 citations):
“S. Kayal and R. V. Ramanujan, "Doxorubicin loaded PVA coated iron oxide nanoparticles for targeted drug delivery," Materials Science and Engineering: C, vol. 30, no. 3, pp. 484-490, 2010/04/06/ 2010, doi: http://dx.doi.org/10.1016/j.msec.2010.01.006”
It was included at the section “2.1.8. Doxorubicin loading” the specific information requested:
“The loading of the water-soluble anticancer drug, (Dox), on the surface, NP-Fe3O4 and NP-Fe3O4/SiO2, was done by the mix of 5 mg of each with 5 of solution (0.1 mg/ in PBS, pH 7.4). This protocol was adapted from the protocol implemented by Kayal, S. et al. [44].The mixtures of nanoparticles and Dox were kept under stirring for 24 h at room temperature. The magnetic nanoparticles were removed by centrifugation at 12000 rpm for 30 min and the supernatant was employed to measure the efficiency rate of drug loading. The Loading efficiency (LE) was determinate by UV-vis spectroscopy, using a T90 from PG Instruments Ltd, taking in to account the absorbance due to the presence of Dox in the solutions at the wavelength of 480 nm. The calculus of the efficiency rate of drug loading was done using the Eq. 1:”
Our results typically showed that the amount of Dox loaded in the magnetic nanoparticle surface was enough to present cytotoxicity in the tests with MCF-7 and reduced cytotoxic effect on health cells (HaCat) (see Figure 4). Dox is well known by the side effects in the body, so it is important to try to minimize the quantity maximizing its toxic effect on the targeted cells.
The amount of Dox loaded in our nanoparticles is comparable with what was described by Kayal S. et al.[1] as well as by Eslami P. et al. [2]. The strategy to use PBS at pH 7.4 worked well and our loading efficiency is higher than mentioned by both authors cited above. The cytotoxic effect of our Dox loaded nanoparticles seems to be slightly higher or similar than what was mentioned by Eslami P.et. all. [2], Norouzi, M. et al. [3] and Siminzar, P. et al. [4], if we consider the maximum amount of Dox present in a suspension with concentration of 10 µg/mL (0.783 µg of Dox) . These data were included in the paper and are highlighted in the new version of manuscript.
5) In the abstract and paper's title, you have to mention about what type of magnetic nanoparticles the material is going. For example, may be better oleic acid coated magnetite
nanoparticle...
Comments to question 5:
As already mentioned above, in our study we carefully analysed the iron oxide nanoparticles (NP-Fe3O4) and silica coated iron oxide (NP-Fe3O4/SiO2). Fe3O4 coated with oleic acid is an intermediary product used to obtain silica coated iron oxide. It is properly described in the experimental part section 2.1.2. Magnetic nanoparticles synthesis. The abstract and title were improved.
6) I have seen zeta potential analysis. Have you measured the sizes of your nanoparticles using dynamic light scattering (DLS)? If yes, the results should be presented in the paper.
Comments to question 6:
Unfortunately the dynamic light scattering experiment was realized in a solvent that makes direct comparison difficult. That is the key reason to DLS data not be added to manuscript.
7) Cell Viability assay:
Why you have not so good results of cell viability (~70%), I mean that you Dox-loaded MNPs are not highly effective? Can you compare the results with other papers and add some
discussion?
Comments to question 7:
Our goal was to adequately prepare magnetic nanoparticles that alone do not present cytotoxic effects on the cells (Figure 3). We also intended to demonstrate the benefits of silica coating on the NP-Fe3O4, which is better evidenced in the case of the HaCat cells. In which, NP-Fe3O4 present some cytotoxicity when exposed to HaCat cells for 24 h, and at this same time point, cells exposed to NP-Fe3O4/SiO2 present higher values of cell viability.
The Dox loaded nanoparticles presented cytotoxic effects, and it is shown in the Figure 4. In addition, it is possible to evidence the specificity of the Dox loaded NPs, in which the cytotoxic effect is higher when exposed to MCF-7 cells. We can highlight the higher cytotoxic effect on these cells in the presence of NP-Fe3O4/SiO2-Dox.
This explanation was improved in the current manuscript at the section 3.7. Cell viability assay. Moreover, some comparisons with the literature were presented.
8) The novelty of the work has to mention in an extensive way in abstract and conclusion part. There are many doxorubicin loaded MNP and MNP-SiO2 for cancer treatment. The references to these works should be presented in the Results and Discussion part.
Comments to question 8:
It was improved in the manuscript.
Minor comments:
line 104 no dot, line 113 two dots, line 102 subscript
NH4OH much more relevant NH3 aq in this solution only 1-2% of NH4+ ions
Why are you present Fig 7 after the Conclusion?
Thank you for your comments!
Figure 7 was placed in the right position at the manuscript.
Reviewer 2 Report
The article entitled “Effects of Magnetic Nanoparticles on Enhance Doxorubicin Treatments in Breast Cancer Cells (MCF-7)” aims to study and investigate the synthesis and characterization of magnetite and silica-coated magnetite nanoparticles and biocompatibility via cellular toxicity tests in terms of cell viability. The results provide insight into the applicability of these magnetic nanoparticles as a drug carrier system. In my opinion, the article sounds interesting but requires revisions prior to being publishable.
1. Abstract is too long and lacks focus.
2. In the Introduction, I suggest adding some references related to Dox encapsulation with other materials. ACS Biomaterials Science & Engineering 3 (10), 2431-2442, Chemical Engineering Journal 383, 123138, Journal of Materials Chemistry B 5 (7), 1507-1517, Chemical Engineering Journal 370, 1188-1199.
3. Moreover, we have several articles coated with silica for dox delivery, I suggest explaining the difference at the end of the introduction.
4. Although coating of silica would address some shortcomings, I suggest explaining how this strategy would be going to overcome the immune clearance. 5. The results were not discussed well. I suggest explaining the reduced loading in silica-coated NPs.
6. It is essential to explore the blood compatibility tests to demonstrate biocompatibility.
7. The authors addressed the release of Dox from NPs but no at least in vitro evidence to support the claim.
8. Moreover, TEM explored the uptake, but I suggest using some fluorescent imaging to demonstrate the biodistribution efficacy.
9. Irregular punctuations and numerous typological (specifically, spacing, superscript, and subscript) errors. I suggest revising the article.
Author Response
Dear Reviewer,
We would like to thank you for the revision on our submitted manuscript ‘‘Doxorubicin loaded magnetic nanoparticles: enhancement of the Doxorubicin effect on breast cancer cell (MCF-7)’’ (magnetochemistry-1869842). The questions were carefully analysed, and the manuscript was modified (improved). Please find below our specific answers.
1) Abstract is too long and lacks focus.
Comments to question 1:
Abstract was improved.
2) In the Introduction, I suggest adding some references related to Dox encapsulation with other materials. ACS Biomaterials Science & Engineering 3 (10), 2431-2442, Chemical Engineering Journal 383, 123138, Journal of Materials Chemistry B 5 (7), 1507-1517, Chemical Engineering Journal 370, 1188-1199.
Comments to question 2:
New references were added to the manuscript and are signalled in the new version.
3) Moreover, we have several articles coated with silica for dox delivery, I suggest explaining the difference at the end of the introduction.
Comments to question 3:
We thank the reviewer for the suggestion. We added comments in the manuscript introduction.
4) Although coating of silica would address some shortcomings, I suggest explaining how this strategy would be going to overcome the immune clearance.
Comments to question 4:
As discussed by Thomas Malachowski and Austin Hassel [5], innate immunity is the first line of defence the body activates upon recognition of foreign invaders, and as a result, it is the first type of resistance nanoparticles encounter. Silica or silica-coated nanoparticles have been studied extensively over the past few decades, and they have gained tremendous traction as drug delivery vessels. In fact, nonporous silica can be problematic as they often invoke immune responses, such as pro-inflammatory cytokines and the recruitment of immune cells, or they can be toxic and disrupt the function of immune cells, eventually leading to apoptosis. Fortunately, additional research demonstrated that porous nanoparticles and surface modifications can suppress these negative side effects and add further functionality to silica and silica-coated nanoparticles.
In this sense, porous silica nanoparticles have been shown to have good biocompatibility, low immunogenicity, and low toxicity in vivo, and they have the capable of acting as “smart” nanoparticles that can have timed-released using a variety of modifications that also assist in target specificity and pre-destination cargo retention. Thus, in our work, we used the strategy to have porous silica-coated nanoparticles, avoiding/reducing the immune effects.
5) The results were not discussed well. I suggest explaining the reduced loading in silica-coated NPs.
Comments to question 5:
The loading procedure was done based in what is related in a well cited paper [1] (more than 400 citations):
“S. Kayal and R. V. Ramanujan, "Doxorubicin loaded PVA coated iron oxide nanoparticles for targeted drug delivery," Materials Science and Engineering: C, vol. 30, no. 3, pp. 484-490, 2010/04/06/ 2010, doi: http://dx.doi.org/10.1016/j.msec.2010.01.006”
It was added at the section “2.1.8. Doxorubicin loading” the information requested:
“The loading of the water-soluble anticancer drug, (Dox), on the surface, NP-Fe3O4 and NP-Fe3O4/SiO2, was done by the mix of 5 mg of each with 5 of solution (0.1 mg/ in PBS, pH 7.4). This protocol was adapted from the protocol used by Kayal, S. et al. [44].The mixtures of nanoparticles and Dox were kept under stirring for 24 h at room temperature. The magnetic nanoparticles were removed by centrifugation at 12000 rpm for 30 min and the supernatant was used to measure the efficiency rate of drug loading. The Loading efficiency (LE) was determinate by UV-vis spectroscopy, using a T90 from PG Instruments Ltd, taking in to account the absorbance due to the presence of Dox in the solutions at the wavelength of 480 nm. The calculus of the efficiency rate of drug loading was done using the Eq. 1:”
Our results showed that the amount of Dox loaded in the magnetic nanoparticle surface was enough to present cytotoxicity in the tests with MCF-7 and low cytotoxic effect on health cells (HaCat) (see Figure 4). Dox is well known by the side effects in the body, so it is important to try to minimize the quantity maximizing its toxic effect on the targeted cells.
The amount of Dox loaded in our nanoparticles are comparable with what was described by Kayal S. et al.[1] as well as by Eslami P. et al. [2]. The strategy to use PBS at pH 7.4 worked well and our loading efficiency is higher than mentioned by both authors cited above. The cytotoxic effect of our Dox loaded nanoparticles seems to be slightly higher or similar than what was mentioned by Eslami P.et. all. [2], Norouzi, M. et al. [3] and Siminzar, P. et al. [4], if we consider the maximum amount of Dox present in a suspension with concentration of 10 µg/mL (0.783 µg of Dox) . These data were included in the paper and are highlighted in the manuscript attached.
6) It is essential to explore the blood compatibility tests to demonstrate biocompatibility.
Comments to question 6:
As addressed by D. F. Williams [6], biocompatibility refers to the ability of a biomaterial to perform its desired function with respect to a medical therapy, without eliciting any undesirable local or systemic effects in the recipient or beneficiary of that therapy, but generating the most appropriate beneficial cellular or tissue response in that specific situation, and optimising the clinically relevant performance of that therapy. In this context, Xiaoming Li et al. [7] taught that nanoparticles toxicity refers to the ability of the particles to adversely affect the normal physiology as well as to directly interrupt the normal structure of organs and tissues of humans and animals. Targeted drug delivery is one of the most intensively explored areas of research and the use of nanoparticles for diagnostic purposes has already entered the biomedical field. In general, iron oxide nanoparticles are classified as biocompatible, showing no severe toxic effects in vitro or in vivo. Studies underline the importance of using different cellular systems for nanotoxicological studies, including primary human cell types. Silica-coated nanoparticles demonstrated a good degree of biocompatibility and enter the cell without affecting cell survival. Based on that we decide do not perform additional biocompatibility tests, as blood compatibility test.
7) The authors addressed the release of Dox from NPs but no at least in vitro evidence to support the claim.
Comments to question 7:
Unfortunately, the in vitro release experiment was not realized. However, the results showed in our work are supported by the literature. They mentioned that the Dox release from magnetic nanoparticles with different coatings, including silica [4, 8] and polymers [3, 9], is pH dependent, been favoured at low pH (around 100 % at pH 4.5). Normally, Dox release experiments are runed at pH 7.4 (blood), pH 5.4 (cellular endosomes) and pH 4.5 (lysosomes), and the release is more effective at low pH. This means that the Dox release will be effective when the nanoparticles are internalized by the cells, particularly cancerous cells, where the internal pH is lower than in health cells. In addition, it can be expected that the cytotoxicity will increase with time due the time of nanoparticles internalization, as well as been recruited into the lysosomes [9]. In our work, it could be observed that the nanoparticles loaded with Dox presented a higher cytotoxic effect on the cancerous cells MCF-7 than in health cells (HaCaT), and add to it, the cytotoxic effect increased with time.
8) Moreover, TEM explored the uptake, but I suggest using some fluorescent imaging to demonstrate the biodistribution efficacy.
Comments to question 8:
Unfortunately, this experiment was not done. We are unable to do so at this time. However, we believe that the lack of this experiment does not affect the conclusions presented in this manuscript.
9) Irregular punctuations and numerous typological (specifically, spacing, superscript, and subscript) errors. I suggest revising the article.
Comments to question 9:
Considering the reviewer comments, we did modifications in our manuscript.
Reviewer 3 Report
The manuscript “Effects of Magnetic Nanoparticles on Enhance Doxorubicin Treatments in Breast Cancer Cells (MCF-7)” by Parcero Hernandes et al. represents an interesting study using magnetic nanoparticles as drug delivery vehicles for doxorubicin.
The whole manuscript is well-written. There are few minor typos and the use of multiple references at some passages should be improved (two references in the same brackets instead of using multiple brackets).
The storyline is very well introduced and the references are balanced. I really like the experimental part, where all methods are well described by the authors.
I do not see any issues with the scientific part. It might not be the most innovative study but the research definitely looks very solid and reproducible.
Abstract:
Can you also introduce abbreviations which you use in the abstract as well?
Materials and Methods:
Can you add how many particles you counted for the histograms?
Results and Discussion:
From the results it is challenging to really exclude other stoichiometric iron oxides and maghemite and therefore the nomenclature magnetite might not be the ideal choice. I would recommend using iron oxide.
Figure 2: The histograms of the TEM images are very small or at least the labelling. May you increase the font size of the labelling or provide the histograms in a supplementary material?
Figure 7 should not be in the conclusion but in the Results and Discussion section.
Author Response
Dear Reviewer,
We would like to thank you for the revision on our submitted manuscript ‘‘Doxorubicin loaded magnetic nanoparticles: enhancement of the Doxorubicin effect on breast cancer cell (MCF-7)’’ (magnetochemistry-1869842). The questions were carefully analysed, and the manuscript was modified (improved). Please find below our specific answers.
Comments:
The whole manuscript is well-written. There are few minor typos and the use of multiple references at some passages should be improved (two references in the same brackets instead of using multiple brackets).
The storyline is very well introduced and the references are balanced. I really like the experimental part, where all methods are well described by the authors.
I do not see any issues with the scientific part. It might not be the most innovative study but the research definitely looks very solid and reproducible.
Abstract:
Can you also introduce abbreviations which you use in the abstract as well?
Materials and Methods:
Can you add how many particles you counted for the histograms?
Results and Discussion:
From the results it is challenging to really exclude other stoichiometric iron oxides and maghemite and therefore the nomenclature magnetite might not be the ideal choice. I would
recommend using iron oxide.
Figure 2: The histograms of the TEM images are very small or at least the labelling. May you increase the font size of the labelling or provide the histograms in a supplementary material.
Figure 7 should not be in the conclusion but in the Results and Discussion section
Answers:
Considering your comments, we did modifications in our manuscript:
- Abstract was improved
- Number of nanoparticles analyzed to build the histograms were at least 250 and this information was added to section “2.6. Transmission Electron microscopy”;
- We thank the reviewer for the suggestion. We agree with this comment. Indeed, with the data presented in the manuscript, it is difficult to conclude in which phase the iron oxide is found. Therefore, we have changed the nomenclature from Fe3O4/magnetite to "iron oxide" throughout the manuscript.
- Figure of TEM was improved (Figure 2 (a) and (b)).
- Figure 7 was located in the wright place.
Round 2
Reviewer 1 Report
I'm a bit confused, the authors haven't changed some comments. I recommend accepting the paper after minor revision.
1) The authors have text error. I have recommended to change it
line 105 C2H5OH subscript
NH4OH much more relevant NH3 aq in this solution only 1-2% of NH4+ ions . I have found four places of this error in the text.
It is very unusual to present any pictures after Conclusions.
2) The authors have improved the references list. However, the new paper about magnetic nanoparticles loaded with doxorubicin should be there. I highly recommend citing recent works 2020-2022 years or maybe some others.
https://www.mdpi.com/search?q=doxorubicine+loaded+iron+oxide+nanoparticles
- High Drug Capacity Doxorubicin-Loaded Iron Oxide Nanocomposites for Cancer Therapy
- Smart Magnetic Nanocarriers for Multi-Stimuli On-Demand Drug Delivery
-Hybrid System for Local Drug Delivery and Magnetic Hyperthermia Based on SPIONs Loaded with Doxorubicin and Epirubicin
-Doxorubicin-Conjugated Iron Oxide Nanoparticles Synthesized by Laser Pyrolysis: In Vitro Study on Human Breast Cancer Cells
3) I the introduction authors present a lot of information about everything, but not about magnetic nanoparticles loaded with doxorubicin. I highly recommend writing several sentences with the references presented above. To present fresh papers and the progress in the area.
4)
The loading procedure was performed based in what is related in a well cited paper [1] (more than 400 citations):
“S. Kayal and R. V. Ramanujan, "Doxorubicin loaded PVA coated iron oxide nanoparticles for targeted drug delivery," Materials Science and Engineering: C, vol. 30, no. 3, pp. 484-490, 2010/04/06/ 2010, doi: http://dx.doi.org/10.1016/j.msec.2010.01.006”
I understand that you can use any paper, but it is 2010 year paper. The drug-loading procedures may go forward for these 12 years.
Of course, you can't do anything with your experiments. It was a recommendation for further work, that Dox-loading may be better. In this way, the concentration of loaded nanoparticles will be lower for your therapeutic effect on the MTT test.
For example, in the paper https://www.mdpi.com/2312-7481/8/5/54/htm
you need only 1.25 µg/mL for 20% cell viability, but in your paper you need not less than 5 µg/mL for Fe3O4/DOX (Fugire 4). For the Fe3O4/SiO2/DOX you require not less than 10 µg/mL.
That is why I have recommended these two works for comparison and to write some words for example, that for better therapeutic effect higher DOX loading amount is required...
- Doxorubicin-Conjugated Iron Oxide Nanoparticles Synthesized by Laser Pyrolysis: In Vitro Study on Human Breast Cancer Cells
- High Drug Capacity Doxorubicin-Loaded Iron Oxide Nanocomposites for Cancer Therapy. In this paper, the authors have synthesized oleic acid coated magnetic nanoparticles with Doxoribicin.
Author Response
Dear Reviewer,
We would like to thank you for the revision on our submitted manuscript ‘‘Doxorubicin loaded magnetic nanoparticles: enhancement of the Doxorubicin effect on breast cancer cell (MCF-7)’’ (magnetochemistry-1869842). The questions were carefully analysed, and the manuscript was modified (improved). Please find below our specific answers.
1) The authors have text error. I have recommended to change it
line 105 C2H5OH subscript
Comment: Thank you for your comment. We will fix that in the final version.
NH4OH much more relevant NH3 aq in this solution only 1-2% of NH4+ ions . I have found four places of this error in the text.
Comment: We could not understand were is the mistake. We used a solution of ammonia in water. It can be denoted by the symbols NH3 aq. Although the name ammonium hydroxide suggests an alkali with composition [NH+4][OH−], it is actually impossible to isolate samples of NH4OH. The ions NH+4 and OH− do not account for a significant fraction of the total amount of ammonia except in extremely dilute solutions. That is the reason why we choose keep the symbols NH3 aq.
It is very unusual to present any pictures after Conclusions.
Thank you for your comment. We will fix that in the final version.
2) The authors have improved the references list. However, the new paper about magnetic nanoparticles loaded with doxorubicin should be there. I highly recommend citing recent works 2020-2022 years or maybe some others.
https://www.mdpi.com/search?q=doxorubicine+loaded+iron+oxide+nanoparticles
- High Drug Capacity Doxorubicin-Loaded Iron Oxide Nanocomposites for Cancer Therapy
- Smart Magnetic Nanocarriers for Multi-Stimuli On-Demand Drug Delivery
-Hybrid System for Local Drug Delivery and Magnetic Hyperthermia Based on SPIONs Loaded with Doxorubicin and Epirubicin
-Doxorubicin-Conjugated Iron Oxide Nanoparticles Synthesized by Laser Pyrolysis: In Vitro Study on Human Breast Cancer Cells
Thank you for your comment. We added these new references.
3) I the introduction authors present a lot of information about everything, but not about magnetic nanoparticles loaded with doxorubicin. I highly recommend writing several sentences with the references presented above. To present fresh papers and the progress in the area.
Thank you for your comment. We added these new references and comments in the introduction.
4)
The loading procedure was performed based in what is related in a well cited paper [1] (more than 400 citations):
“S. Kayal and R. V. Ramanujan, "Doxorubicin loaded PVA coated iron oxide nanoparticles for targeted drug delivery," Materials Science and Engineering: C, vol. 30, no. 3, pp. 484-490, 2010/04/06/ 2010, doi: http://dx.doi.org/10.1016/j.msec.2010.01.006”
I understand that you can use any paper, but it is 2010 year paper. The drug-loading procedures may go forward for these 12 years.
Of course, you can't do anything with your experiments. It was a recommendation for further work, that Dox-loading may be better. In this way, the concentration of loaded nanoparticles will be lower for your therapeutic effect on the MTT test.
For example, in the paper https://www.mdpi.com/2312-7481/8/5/54/htm
you need only 1.25 µg/mL for 20% cell viability, but in your paper you need not less than 5 µg/mL for Fe3O4/DOX (Fugire 4). For the Fe3O4/SiO2/DOX you require not less than 10 µg/mL.
That is why I have recommended these two works for comparison and to write some words for example, that for better therapeutic effect higher DOX loading amount is required...
- Doxorubicin-Conjugated Iron Oxide Nanoparticles Synthesized by Laser Pyrolysis: In Vitro Study on Human Breast Cancer Cells
- High Drug Capacity Doxorubicin-Loaded Iron Oxide Nanocomposites for Cancer Therapy. In this paper, the authors have synthesized oleic acid coated magnetic nanoparticles with Doxoribicin.
Thank you for your comment. We added some comparisons using the recommended papers.